# Zero-Shot Image Classification Based on a Learnable Deep Metric

**DOI:** 10.3390/s21093241

**Published:** 2021-05-07

**Authors:** Jingyi Liu, Caijuan Shi, Dongjing Tu, Ze Shi, Yazhi Liu

**Affiliations:** College of Information Engineering, North China University of Science and Technology, Tangshan 063210, China; godjing12138@163.com (J.L.); t1176242923@163.com (D.T.); 547143680@163.com (Z.S.); liuyazhi@ncst.edu.cn (Y.L.)

**Keywords:** zero-shot learning, deep metric, common space embedding, relation network, image classification, deep learning

## Abstract

The supervised model based on deep learning has made great achievements in the field of image classification after training with a large number of labeled samples. However, there are many categories without or only with a few labeled training samples in practice, and some categories even have no training samples at all. The proposed zero-shot learning greatly reduces the dependence on labeled training samples for image classification models. Nevertheless, there are limitations in learning the similarity of visual features and semantic features with a predefined fixed metric (e.g., as Euclidean distance), as well as the problem of semantic gap in the mapping process. To address these problems, a new zero-shot image classification method based on an end-to-end learnable deep metric is proposed in this paper. First, the common space embedding is adopted to map the visual features and semantic features into a common space. Second, an end-to-end learnable deep metric, that is, the relation network is utilized to learn the similarity of visual features and semantic features. Finally, the invisible images are classified, according to the similarity score. Extensive experiments are carried out on four datasets and the results indicate the effectiveness of the proposed method.

## 1. Introduction

Thanks to the development of deep learning models, image classification and image recognition have made continuous progress. However, most of existing deep learning models [1,2,3,4] are supervised and they can only classify and recognize seen classes with labeled samples. The only way to recognize the novel classes is to retrain the classifiers with a large number of novel, labeled samples. To classify and recognize unseen classes, human reasoning process is utilized to simulate the unseen classes in human brains, reading the description of objects and recognizing them. Similarly, zero-shot learning is proposed and it makes the deep learning models have the ability to reason similar to humans, classify and recognize the new classes, even which have never been seen before.

Zero-shot learning (ZSL) has become a new direction derived from transfer learning and its training and testing samples are independent and disjointed. ZSL aims to replace the low-dimensional features of samples with high-dimensional semantic features so that the trained model has the ability to transfer.

In recent years, a kind of typical ZSL methods based on space embedding [5,6,7] uses the correlation between seen and unseen classes to complete the attribute transferring from seen classes to unseen classes. According to the mapping of vision and semantic, space embedding methods [8,9,10] can be divided into three categories: semantic space embedding methods, visual space embedding methods, and common space embedding methods. As a typical semantic space embedding method, the Semantic AutoEncoder (SAE) model [8] maps visual features to semantic space directly. But there is a hubness problem, that is, the recognition results of unseen classes are biased toward seen classes, due to the lack of visual features of unseen classes. As one visual space embedding method, the Deep Embedding Model (DEM) maps semantic features into the visual space [9]. Although DEM can alleviate the hubness problem to a certain extent, the inconsistency between the manifold of visual features and semantic features leads to the semantic gap. Consequently, common space embedding methods are proposed to map visual features and semantic features to the same embedding space for achieving good classification performance. For example, structured joint embedding (SJE) [10] is one common space embedding method. Therefore, to alleviate the semantic gap problem, we propose a new ZSL method based on the common space embedding in this paper.

In order to learn the relationship between visual features and semantic features easily, exiting ZSL methods [8,9,10,11,12,13,14] usually use the nearest neighbor search methods with predefined fixed measures. Ji et al. [11] have adopted attribute similarity to constrain the distance between categories in the same modality, meanwhile, hash codes have been generated according to the category similarity and attribute similarity to perform the approximate nearest neighbor (ANN) search. In [12], the labels have been sorted in depth according to the distance and then the ranking SVM has been directly used to perform zero-shot multi-label prediction. Sandouk et al. [13] have used the Euclidean distance between embedded concepts in the concept embedding space to reflect the semantic similarity; while the simple metric has the limitation of unlearnable and being predefined in advance. To overcome these limitations, Sung et al. [14] have proposed the relation network model (RN) to learn a learnable end-to-end deep metric for comparing the relation between visual features and semantic features with the relationship scores. Inspired by the relation network model, we propose a new ZSL method based on the learnable deep metric in this paper. 

Therefore, we propose a new zero-shot image classification method based on a learnable deep metric (ZIC-LDM). ZIC-LDM model is composed of the common space embedding module and the relation module. The common space embedding module is adopted to map the visual features and semantic features into a common space and the relation module is to calculate the similarity score between the visual features and semantic features by using the end-to-end learnable deep metric to achieve good relationship matching. Our proposed method, ZIC-LDM, can learn the correlation between visual features and semantic features in the common space with the learnable deep metric, and it adjusts the correlation end-to-end in a data-driven way. This can greatly alleviate the semantic gap problem caused by the inconsistency between the manifold of visual features and semantic features. ZIC-LDM is applied to the traditional zero-shot image classification task and the generalized zero-shot image classification task, respectively. Experiments are conducted on widely used datasets and the experimental results indicate that ZIC-LDM has the ability to achieve better zero-shot image classification performance compared with other methods.

## 2. Related Work

### 2.1. Zero-Shot Learning

ZSL relies on the labeled seen classes and the semantic information associated with unseen classes and seen classes. In the early stage, the ZSL methods, such as Direct Attribute Prediction (DAP) and Indirect Attribute prediction (IAP) [15], predict the testing samples by training an attribute classifier. With the development of ZSL, current ZSL methods mainly include two categories: space embedding methods and generative model methods. The space embedding methods [8,9,10] rely on an embedding space, in which the attribute migration from seen classes to unseen classes is completed. The generative model methods utilize different generative models, such as generative adversarial networks (GANs) [16,17], variational autoencoders (VAEs) [18,19], and flow-based models (Flows) [20] to directly generate visual features of unseen classes, and then transform the zero-shot learning problem into a traditional supervised learning problem. For example, [16] has proposed a triple discriminator GAN (TDGAN), which employs a GAN with three discriminators to synthesize visual features for images of unseen classes. Ref. [17] has proposed a multi-modal generative adversarial network (M2GAN) to fuse various types of class semantic prototypes, which are achieved in an adversarial framework. Machot et al. [21] have designed a ZSL algorithm by exploiting heterogeneous knowledge between sensor data and semantic space, and then they have spread this algorithm from recognizing unseen classes to unseen human action. Matsuki et al. [22] have proposed an extended word vector-based algorithm by analyzing several ZSL results of embedding semantic features in semantic space. Ohashi et al. [23] have considered that different classes might exist some same attributes, which would influence the classification. Therefore, they have proposed one method to calculate the importance of every attribute of each class.

Though the performance of ZSL has been constantly improved, the setting of the testing stage is too strict and it cannot truly reflect the scene of object recognition in the real world. Therefore, Chao et al. [24] have proposed the generalized zero-shot learning (GZSL) to be closer to the reality of testing stage, considering the amount of seen classes is far more than that of unseen classes. That is, the testing data come from unseen classes and seen classes in GZSL, while the testing data come from unseen classes in ZSL.

### 2.2. Meta Learning

Meta learning is to learn new tasks by using prior knowledge and the known experience. The existing meta methods can be roughly divided into three categories: learning to fine-tune based methods, RNN memory-based methods, and metric learning-based methods. Learning to fine-tune based method [25] learns an initial parameter first, and then it only needs a few samples for training to solve new problems through several gradient descent steps. The RNN memory-based methods use memory recurrent neural networks for meta learning. For example, an external memory model has been used to solve the problem of few samples [26]. The metric learning-based method [27] maps the input space (such as a picture) to a new embedding space by learning an embedding function, and then uses similarity measures (such as Cosine distance and Euclidean distance) to distinguish individual classes. The relation network model (RN) [14] classifies the images of new classes by calculating the relationship scores between the query image and several examples of each new class.

### 2.3. Semantic Features

Semantic features, representing various details of categories, such as characteristics of animals, human behaviors, and scene descriptions, are used to distinguish different objects in ZSL. In general, semantic features are usually divided into three categories: user-defined attribute features, word vectors, and text features. The most common category of semantic features is the user-defined attribute features, which are the specific description of a certain category. The user-defined attribute features have been used to construct semantic space for improving the accuracy of zero-shot classification [28]. Word vectors (Word2Vec) are converted from a text by natural languages processing technology, such as CBOW [29], Skip-gram [29], and GloVe [30]. Text features are transformed from the text description of a certain category through text coding models.

### 2.4. Similarity Measure for Zero-Shot Image Classificaiton

Most zero-shot image classification methods project features extracted by deep learning network into the embedding space. An end-to-end deep learning model could learn better embedding space and a more flexible model. However, these deep models estimate loss through different loss functions. Socher et al. [31] have employed Euclidean distance to match features and attributes in a simple way. Xie et al. [32] have considered the compatibility loss, which has advantages to learn local features. Ref. [33] has exploited the Margin-based loss and has integrated the language model into a neural network, and then increase the separability of features by the end-to-end learning. Ba et al. [34] have considered the binary cross-entropy loss, hinge loss, and Euclidean distance loss simultaneously to predict testing samples through text corpus. DEM [9] has exploited the least square loss, which could not only jointly learn the language model and the embedding space but also fuse text description and multiple modal data. RN [14] has exploited the MSE loss to calculate the similarity score between visual features and semantic attributes to classify unseen classes.

## 3. Methodology

### 3.1. Task Define

In the zero-shot image classification task, the seen classes $S={\left\{\left(x_{p},y_{p}\right)\right\}}_{p=1}^{n_{s}}$ are taken as the training set, where $x_{p}\in X_{S}$ is the p-th image of the seen classes and $y_{p}\in Y_{S}$ is the corresponding class label. The unseen classes $U={\left\{(x_{q},y_{q}\right\}}_{q=1}^{n_{u}}$ serve as the testing set, where $y_{q}\in Y_{U}$ is the corresponding class label. Seen classes and unseen classes constitute the whole dataset, but they don’t intersect: $X_{S}\cup X_{U}=X$, $Y_{S}\cup Y_{U}=Y$, and $Y_{S}\cap Y_{U}=\emptyset$. In our paper, we adopt the user-defined attribute features as the semantic features v. $v_{c}$ and $v_{d}$ respectively represent the semantic features of seen classes and unseen classes with the number of classes c and d. In the testing stage, for testing sample $x_{q}$ and semantic features $v_{d}$, the purpose of zero-shot image classification is to predict the corresponding $y_{q}$ for $x_{q}$.

### 3.2. Model

In this paper, a zero-shot image classification framework, based on learnable deep metric ZIC-LDM is proposed. Figure 1 shows the framework of ZIC-LDM, which is mainly composed of two modules: the relation module and the common space embedding module. In the common space embedding module, the visual features of a given image x are extracted as $f_{\varphi }\left(x\right)$ by using the residual network ResNet101. Then $f_{\varphi }\left(x\right)$ is mapped into the common space through a fully connected layer and now the visual features are defined as $g_{\varphi }\left(x\right)$. In addition, two fully connected layers are used to map the semantic features v to the same common space, where the semantic features are expressed as $h_{\varphi }\left(v\right)$. In the relation module, the visual features $g_{\varphi }\left(x\right)$ and the semantic features $h_{\varphi }\left(v\right)$ are first concatenated, and then the similarity score is calculated by two fully connected layers in a data-driven way. At last, image classification can be completed according to the similarity score.

#### 3.2.1. Common Space Embedding Module

Common space embedding module maps visual features and semantic features to the same common space.

First, the visual features $f_{\varphi }\left(x\right)$ of a given image x can be obtained by using the residual network ResNet101 with the parameter $W_{f}$. The visual features $f_{\varphi }\left(x\right)$ can be expressed as:(1)$$f_{\varphi }\left(x\right)=W_{f}\times x$$

Then, the visual features $f_{\varphi }\left(x\right)$ is mapped to the common space through a fully connected layer with the parameter $W_{g}$, and the visual features in common space can be expressed as $g_{\varphi }\left(x\right)$:(2)$$g_{\varphi }\left(x\right)=W_{g}\times f_{\varphi }\left(x\right)$$

For semantic features v, which are mapped to the same common space through two fully connected layers with the parameter. At this time, the semantic features in the common space are expressed as $h_{\varphi }\left(v\right)$:(3)$$h_{\varphi }\left(v\right)=W_{h}\times v$$

#### 3.2.2. Relation Module

Relation module $r_{\omega }$ realizes zero-shot image classification by calculating the similarity score of visual features and semantic features. After visual features and semantic features are embedded into the common space and connected, the relation module calculates a scalar between 0 and 1 according to the parameters of the relation network to represent the learned relationship between visual features and semantic features in the relation module, which referred to the similarity score. This scalar refers to as the similarity score. The higher the similarity score is, the more matching visual features and semantic features are. First, the visual features and semantic features are concatenated followed by a RELU function activated fully connected layer and a sigmoid function activated fully connected layer in turn, and finally the similarity score is calculated. In the training stage, the visual features $g_{\varphi }\left(x_{p}\right)$ of the image and the semantic features $h_{\varphi }\left(v_{c}\right)$ obtained in the common space embedding module are concatenated, and then the similarity score $s_{p,m}$ is calculated after two full connection layers. The similarity score $s_{p,m}$ can be expressed as:(4)$$s_{p,m}=r_{\omega }\left(C\left(g_{\varphi }\left(x_{p}\right),h_{\varphi }\left(v_{c}\right)\right)\right),m=1,2,\ldots ,c$$
In the above Equation (4), C(,) represents the operation of concatenation.

Here we expect a regression problem to calculate the similarity score with the learnable deep metric. The similarity score is a concrete value in the range of {0, 1}. However, in order to avoid restrictions, we approximately regard it as a binary classification problem. When visual features and semantic features match, the similarity score is 1, otherwise, the similarity score is 0.

#### 3.2.3. Objective Function

In this paper, mean square error (MSE) is used as the loss function and it can be calculated with the similarity score $s_{p,m}$ and the real category label $y_{\left(v_{c}\right)}$ of the seen class. The loss function is can be expressed as follow:(5)$$L=\sum_{p=1}^{n_{s}}\sum_{m=1}^{c}\left(s_{p,m}-1\left(y_{p}\equiv y_{\left(v_{c}\right)}\right)\right)$$

To make the relation module match the visual features and semantic features belonging to the same category well, the proposed ZIC-LDM needs to be trained by minimizing Formula (5):(6)$$W_{g}^{*},W_{h}^{*},r_{\omega }^{*}\leftarrow \underset{W_{g},W_{h},r_{\omega }}{\mathrm{argmin}}L$$

#### 3.2.4. Model Implementation

The training process of ZIC-LDM, i.e., zero-shot image classification model based on a learnable deep metric, is shown in Algorithm 1.
**Algorithm 1:** Training process of ZIC-LDM **Input:** Training process iteration rounds epochs, batch size m, learning rate lr, semantic features v, FC parameter $W_{h}$ initialized for semantic features mapping, visual feature $f_{\varphi }\left(x\right)$, FC parameter $W_{g}$ for visual features mapping and relation module $r_{\omega }$. **Output:** Optimized FC parameter $W_{h}^{*}$ for semantic features mapping, FC parameter $W_{g}^{*}$ for visual features mapping and relation module $r_{\omega }^{*}$.1**for** epoch=0,1,2,…, epochs−1**do**2  **for** i=0,1,2,…, ntrain/m−1
**do**3   Sampling m training samples x and corresponding label y from seen classes;4   Mapping $f_{\varphi }\left(x\right)$ into common space: $g_{\varphi }\left(x\right)\leftarrow W_{g}\times f_{\varphi }\left(x\right)$;5   Mapping v into common space: $h_{\varphi }\left(v\right)\leftarrow W_{h}\times v$;6   Concatenate $g_{\varphi }\left(x\right)$ and $h_{\varphi }\left(v\right)$;7   Calculate similarity score: $s_{p,m}\leftarrow r_{\omega }\left(g_{\varphi }\left(x\right),h_{\varphi }\left(v\right)\right)$;8   Calculate MSE loss: $L=MSE\left(s_{p,m},y_{v}\right)$;9   Update FC parameters for semantic features mapping, FC parameters for visual features mapping and relation module:   $W_{g}^{*},W_{h}^{*},r_{\omega }^{*}\leftarrow Adan\left(\nabla W_{g},W_{h},r_{\omega }[L],W_{g},W_{h},r_{\omega },lr\right)$;10  **end for**11**end for**

### 3.3. Testing Process

In this chapter, zero-shot image classification task and generalized zero-shot image classification task are tested respectively.

#### 3.3.1. Zero-Shot Image Classification

In the zero-shot image classification task, for a given image $x_{q}\in x_{U}$ of the unseen class, the visual features $f_{\varphi }\left(x_{q}\right)$ are extracted and then they are mapped to the common space with the representation $g_{\varphi }\left(x_{q}\right)$ by the trained fully connected layer FC3. Then, the fully connected layers FC1 and FC2 are used to map the semantic features $v_{d}$ of the unseen class into same common space to obtain the semantic features $h_{\varphi }\left(v_{d}\right)$ of the unseen classes (*d* is the number of unseen classes). $h_{\varphi }\left(v_{d}\right)$ and $g_{\varphi }\left(x_{q}\right)$ are concatenated, followed by the calculation of their similarity, namely the similarity score:(7)$$s_{q,m}=r_{\omega }\left(C\left(g_{\varphi }\left(x_{q}\right),h_{\varphi }\left(v_{d}\right)\right)\right),m=1,2,\ldots ,d$$

Finally, the class with the highest similarity score is taken as the prediction label. This can be expressed as:(8)$$\tilde{Y}=\arg\max s_{q,m}$$

#### 3.3.2. Generalized Zero-Shot Image Classification

When the generalized zero-shot classification is carried out, the testing classes include both seen classes and unseen classes, that is $X=X_{S}\cup X_{U}$. At this time, the visual features $f_{\varphi }\left(x\right)$ are expressed as $g_{\varphi }\left(x\right)$ in common space. The semantic features mapped to the common space are $h_{\varphi }\left(v\right)$, where $v=v_{c}\cup v_{d}$. The matching degree of visual features and semantic features of the image, i.e., the similarity score s, can be calculated as follow:(9)$$s=r_{\omega }\left(C\left(g_{\varphi }\left(x\right),h_{\varphi }\left(v\right)\right)\right)$$

The class with the highest similarity score is taken as the label of the prediction. This is expressed as:(10)$$\tilde{Y}=\arg\max s$$

## 4. Experiments

The proposed method ZIC-LDM is tested and compared with several existing methods on four datasets. Experiments are conducted and the results indicate the effectiveness of ZIC-LDM.

### 4.1. Dataset and Settings

In our experiments, four datasets commonly used in zero-shot image classification are selected: Animals with Attributes 1 and 2 (AwA1 [15] and AwA2 [35]), CUB (CUB-200–2011) [36], and SUN (SUN Attribute) [37]. AwA1 and AwA2 contain 30,745 and 37,322 animal images of 50 categories, respectively, of which 40 categories are training classes and 10 categories are testing classes. CUB contains a total of 11,788 images of 200 bird species with 150 training classes and 50 testing classes. SUN contains 14,340 images of 717 categories, of which 645 categories are training classes and 72 categories are testing classes. In terms of semantic features, AwA1 and AwA2 use 85-dimension semantic features. For CUB and SUN, 312-dimension and 102-dimension semantic features are used respectively. The semantic features used are the user-defined attribute features, which are provided directly by the datasets.

For the common space embedding module, the pooling layer of the top layer of RerNet101 without fine-tuning is used to extract the visual features $f_{\varphi }\left(x\right)$, whose dimension is 1024. MLP network is used to learn the semantic features $h_{\varphi }\left(v\right)$. For the relation module, $g_{\varphi }\left(x\right)$ and $h_{\varphi }\left(v\right)$ are concatenated and then the relationship between the visual features and semantic features of the image in the common space is calculated through two fully connected layers FC4 and FC5. In zero-shot image classification, the hubness problem often occurs in cross-modal mapping, so *L_2_* regularization is added to FC1 and FC2 at the fully connected layers to alleviate this problem. Besides, our framework is trained in the embedded network with a weight decay of 10^−5^, and the Adam algorithm is used to initialize the learning rate to 10^−5^.

To verify the effectiveness of the proposed ZIC-LDM, we compare it with the following 13 models: Direct Attribute Prediction (DAP) [15], Convex combination of Semantic Embeddings (ConSE) [38], Embarrassingly Simple Zero-Shot Learning (ESZSL) [7], Attribute Label Embedding (ALE) [39], Synthesized Classifiers for zero-shot Learning (SynC) [40], Semantic Auto Encoder (SAE) [8], Relation Network (RN) [14], Structured Joint Embedding (SJE) [10], Cross-Class Sample Synthesis (CCSS) [41], Gaussian [42], Simple and Effective Localized Attribute Representations (SELAR) [43], Modeling Inter and Intra-Class Relations (MIIR) [44], Marginalized Latent Semantic Encoder (MLSE) [45]. SJE is taken as the baseline.

### 4.2. Traditional Zero-Shot Image Classification

Top-1 accuracy is usually used as the criterion for the image classification. The prediction is accurate when the predicted class is correct. Averaging the accuracies of all images can have a good effect on the classes with dense images. What is more, for some classes with relatively rare images, the average values of each group of correct predictions are calculated, respectively, that is, the average top-1 accuracy of each class is measured. In the traditional zero-shot image classification, the mean precision of top-1 is adopted as the criterion. The experimental results on AwA1, AwA2, CUB, and SUN datasets are shown in Table 1, and the best results are in bold.

As can be seen from Table 1:The results of ZIC-LDM on AwA1, AwA2, CUB, and SUN datasets are better than those of the baseline SJE with the increase of 4.0%, 5.8%, 2.9%, and 5.2%, respectively. In addition, compared with the latest models Gaussian and SELAR, ZIC-LDM also achieves excellent results, which shows that our proposed model is effective in zero-shot image classification. Therefore, the ZIC-LDM with the learnable deep metric can learn good visual-semantic relationship.Compared with the methods, such as DAP, CONSE, ESZSL, ALE, and SynC, which use the predefined fixed metrics, ZIC-LDM has achieved the best results on AwA1, AwA2, CUB, and SUN datasets. This indicates that the learnable deep metric makes ZIC-LDM learn the visual-semantic relationship well.Compared with the method SAE based on semantic space embedding and methods RN and CCSS based on visual space embedding, ZIC-LDM based on common space embedding has achieved the best results on AwA1, AwA2, CUB, and SUN datasets. It shows that common space embedding can alleviate the semantic gap problem.

In addition, the confusion matrices and visualization results of ZIC-LDM on AwA1 and AwA2 datasets are given in Figure 2 and Figure 3. It can be seen from the diagonal values of the confusion matrices in Figure 2 that ZIC-LDM can accurately recognize most unseen classes. Figure 3 shows the visualization distribution of 10 unseen classes with t-SNE. The distribution of same class is more concentrated, and the distribution of different classes is more dispersed. Figure 2 and Figure 3 show the feasibility and effectiveness of the learnable deep metric and the common space embedding.

All in all, the above results indicate that our proposed method ZIC-LDM can obtain good zero-shot image classification performance. This is mainly due to the common space embedding module and relation module used in ZIC-LDM. The learnable deep metric helps ZIC-LDM learn the relationship between visual features and semantic features and the common space embedding module alleviates the semantic gap problem.

### 4.3. Generalized Zero-Shot Image Classification

In the generalized zero-shot image classification, the harmonic mean is selected as the evaluation criterion to make seen classes and unseen classes both have high accuracy. Firstly, the average top-1 accuracy per class of seen classes and unseen classes is calculated, and then the harmonic mean of seen classes and unseen classes is computed. The expression of the harmonic mean is as follow:(11)$$H=\frac{2\times \left(S\times U\right)}{\left(S+U\right)}$$
where S represents the average top-1 precision of seen classes, and U represents the average top-1 precision of unseen classes. In the process of model training, seen class samples may make the model over-fit, resulting in an imbalance between the accuracy of seen class samples and the accuracy of unseen class samples. So, U and H are the main evaluation criterion of generalized zero-shot image classification.

Our proposed method ZIC-LDM is compared with other 12 methods in the generalized zero-shot image classification task on AwA1, AwA2, CUB, and SUN datasets. The results are shown in Table 2 and the optimal results are in bold.

It can be seen from Table 2 that: Compared with the baseline SJE, the average top-1 accuracy U of ZIC-LDM is improved by 21.4%, 23.9%, 16.8%, and 8.8% respectively on four datasets. The average harmonic mean H of ZIC-LDM is also superior to that of SJE, with the increasements of 28.4%, 33.3%, 15.5%, and 7.8%, respectively. In addition, compared with other traditional methods, i.e., DAP, SynC, ESZSL, ALE, SAE, and Gaussian, ZIC-LDM obtains the optimal U and H on AwA1, AwA2, CUB, and SUN datasets. This indicates that ZIC-LDM has more advantages in solving the deviation problem of unseen class.Compared with the learnable deep metric-based method RN, the U of ZIC-LDM is improved by 1.3%, 1.9%, and 2.2% on AwA1, AwA2, and CUB. The average harmonic mean H of ZIC-LDM is improved by 1.3%, 2.1%, and 2.1%, respectively. This indicates that common space embedding can relieve the semantic gap in generative zero-shot learning.Compared with the latest methods MLSE, MIIR, and SELAR, ZIC-LMD has the best U and H on AwA2, CUB, and SUN datasets. It shows that the combination of learnable deep metric and common space embedding is advanced and effective in generalized zero-shot image classification task. 

In general, ZIC-LDM achieves good result in generative zero-shot learning. It benefits from learning the relationship between visual and semantic features by using the learnable deep metric and using common space embedding to relieve the semantic gap.

### 4.4. Loss Convergence Analysis

The essence of the semantic gap problem is that the model cannot learn the good mapping relationship between visual features and semantic features because of their manifold differences. In the process of model training, the convergence of the loss function can determine whether the semantic gap problem is alleviated. The loss convergence of ZIC-LDM is analyzed on AwA1 and AwA2 datasets, respectively. In these experiments, ZIC-LDM is compared with RN, which is based on the learnable deep metric and the visual embedding space. The loss of the first 2000 iterations is selected as the comparison, and the loss output per 400 iterations is taken as the record. The loss convergence curves on AwA1 dataset and AwA2 dataset are shown in Figure 4.

From Figure 4 we can see that ZIC-LDM has good convergence. Compared with RN, the common embedding space adopted in ZIC-LDM makes it alleviate the semantic gap problem between visual features and semantic features, and then makes its loss converge faster than that of RN. It shows that common space embedding alleviates the semantic gap problem.

In addition, the computing time of ZIC-LDM is relatively short. For example, our ZIC-LDM method takes about 125 s or 1500s with 25,000 epochs or 300,000 epochs to perform traditional and generalized zero-shot image classification, respectively, on 30,745 images of 50 categories on the AwA1 dataset, and the experimental results are good. We use NVIDIA GTX1080Ti, made by Micro Star International (Shenzhen, China), and PyTorch1.0 as the testing environment in our experiments.

### 4.5. Distance Metric Study

In this section, we conduct experiments on distance metrics study. Three predefined fixed distanced measures, Euclidean distance (ED), Cosine similarity (CS), and Mahalanobis metric learning (MML) are compared with the learnable deep metric used in our method ZIC-LDM. The experimental results are shown in Table 3 and the optimal results are in bold, where T is top-1 accuracy for traditional zero-shot image classification.

From Table 3, we can see that the learnable deep metric obtains the best results compared with the predefined fixed distanced measures learnable deep metric ED, CS, and MML. This is attributed to its ability to learn the relationship between visual features and semantic features.

## 5. Conclusions

In this paper, the zero-shot image classification method based on learnable deep metric ZIC-LDM is proposed to overcome the limitation of similarity between visual features and semantic features of images and alleviate the semantic gap problem by using the relation module and the common space embedding module. The relation module uses the learnable deep metric to learn the good visual-semantic relationship, as well as the common space embedding module can learn the correlation between visual features and semantic features in the common space to alleviates the semantic gap problem. Compared with other methods, especially the baseline SJE, the proposed method has better performance in both the traditional ZSL task and the GZSL task on four datasets. This indicates the effectiveness and advancement of the proposed method ZIC-LDM. However, some categories with low correlation (i.e., birds and cars) have a limit when the learnable deep metric learns the relationship between visual features and semantic features in the knowledge transforming process, due to user annotated semantic features. Then our future research will exploit the learnable deep metric to learn visual-semantic relationships in graph neural networks, according to feature nodes of different categories. In addition, we will also research the domain shift problem in our future work.

## Figures and Tables

**Figure 1 sensors-21-03241-f001:**
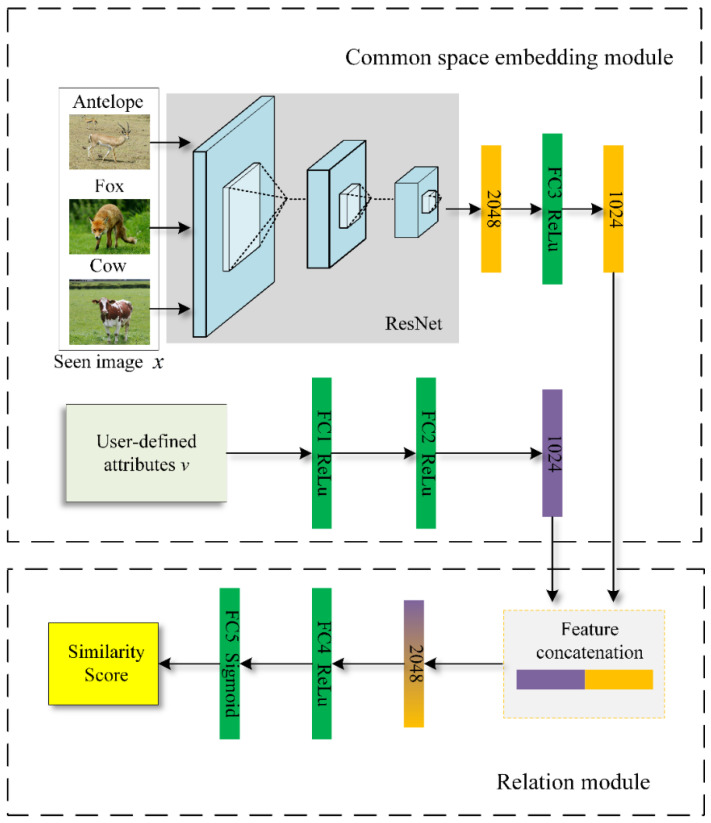
The framework of ZIC-LDM.

**Figure 2 sensors-21-03241-f002:**
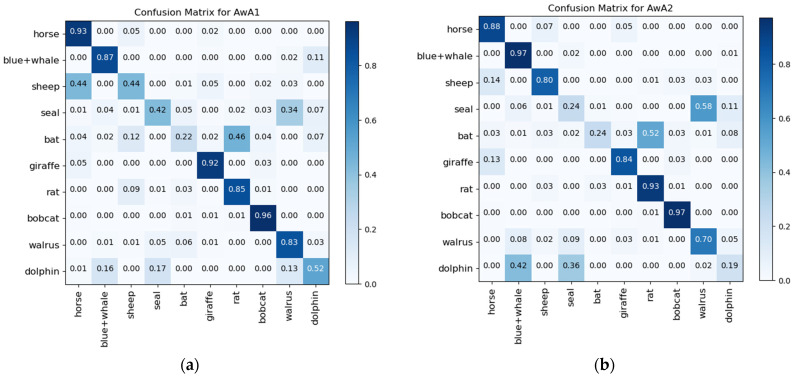
Confusion matrices of ZIC-LDM on AwA1 and AwA2 datasets respectively. (**a**) is confusion matrix for AwA1 and (**b**) is confusion matrix for AwA2.

**Figure 3 sensors-21-03241-f003:**
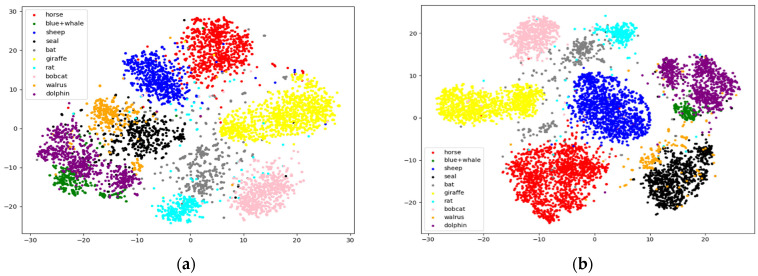
Visualization of the distribution of the 10 unseen class images in the common embedding space on AwA1 and AwA2 using t-SNE. (**a**) is t-SNE result on AwA1 and (**b**) is on AwA2.

**Figure 4 sensors-21-03241-f004:**
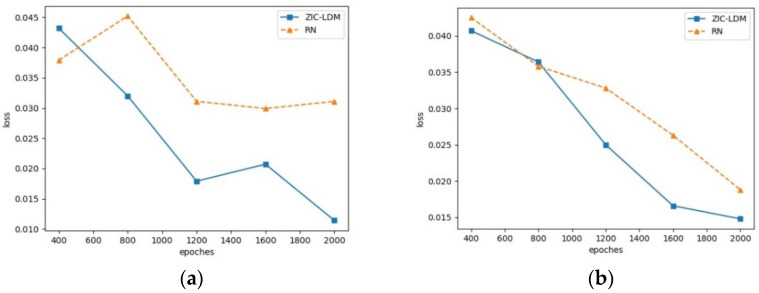
Loss convergence curves on AwA1 and AwA2 datasets. (**a**) is the loss convergence curves on AwA1 dataset and (**b**) is the loss convergence curves on AwA2.

**Table 1 sensors-21-03241-t001:** Accuracy of models for zero-shot learning (%).

Model	AwA1	AwA2	CUB	SUN
DAP [15]	44.1	46.1	40.0	39.9
ConSE [38]	45.6	44.5	34.3	38.8
ESZSL [7]	58.2	58.6	53.9	54.5
ALE [39]	59.9	62.5	54.9	58.1
SynC [40]	54.0	46.6	55.6	56.3
SAE [8]	53.0	54.1	33.3	40.3
CCSS [41]	56.3	63.7	44.1	56.8
Gaussian [42]	60.5	61.2	52.1	58.7
SELAR [43]	-	66.7	56.4	57.8
RN [14]	68.2	64.2	55.6	-
SJE [10]	65.6	61.9	53.9	53.7
ZIC-LDM	**69.6**	**67.7**	**56.8**	**58.9**

**Table 2 sensors-21-03241-t002:** Accuracy of models for generalized zero-shot learning (%).

Model	AwA1			AwA2			CUB			SUN		
	U	S	H	U	S	H	U	S	H	U	S	H
DAP [15]	0.0	88.7	0.0	0.0	84.7	0.0	1.7	67.9	3.3	4.2	25.1	7.2
SynC [40]	8.9	87.3	16.2	10.0	90.5	18.0	11.5	**70.9**	19.8	7.9	**43.3**	13.4
ESZSL [7]	6.6	75.6	12.1	5.9	77.8	11.0	12.6	63.8	21.0	11.0	27.9	15.8
ALE [39]	16.8	76.1	27.5	14.0	81.8	23.9	23.7	62.8	34.4	21.8	33.1	26.3
SAE [8]	1.8	77.1	3.5	1.1	82.2	2.2	7.8	54.0	13.6	8.8	18.0	11.8
ConSE [38]	0.4	88.6	0.8	0.5	90.6	1.0	1.6	72.2	3.1	6.8	39.9	11.6
Gaussian [42]	6.1	81.3	11.4	7.3	79.1	13.3	17.5	59.9	27.1	18.2	33.2	23.5
MLSE [45]	-	-	-	23.8	83.2	37.0	22.3	71.6	34.0	20.7	36.4	26.4
MIIR [44]	-	-	-	17.6	87.0	28.9	30.4	65.8	41.2	22.0	34.1	26.7
SELAR [43]	-	-	-	31.6	80.3	45.3	32.1	63.0	42.5	22.8	31.6	26.5
RN [14]	31.4	**91.3**	46.7	30.0	**93.4**	45.3	38.1	61.1	47.0	-	-	-
SJE [10]	11.3	74.6	19.6	8.0	73.9	14.4	23.5	59.2	33.6	14.7	30.5	19.8
ZIC-LDM	**32.7**	90.5	**48.0**	**31.9**	92.5	**47.4**	**40.3**	62.9	**49.1**	**23.5**	33.9	**27.6**

**Table 3 sensors-21-03241-t003:** Distance Metric Study (%).

Model	AwA1				AwA2				CUB			
	T	U	S	H	T	U	S	H	T	U	S	H
ED	55.2	5.4	68.3	10.0	55.8	5.7	69.5	10.5	42.7	8.2	53.1	14.2
CS	55.4	5.9	68.6	10.9	55.7	5.1	70.2	9.5	42.9	8.5	53.5	14.7
MML	56.7	6.3	70.4	11.6	56.7	6.1	73.7	11.3	16.8	10.5	54.1	17.6
ZIC-LDM	**69.6**	**32.7**	**90.5**	**48.0**	**67.7**	**31.9**	**92.5**	**47.4**	**56.8**	**40.3**	**62.9**	**49.1**

## Data Availability

Not applicable.

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
