# Peer review of "Zero-Shot Image Classification Based on a Learnable Deep Metric"

_sensors, 2021, doi:10.3390/s21093241_

Round 1
Reviewer 1 Report
In this paper, a zero-shot image classification method is proposed, in which the seen images and unseen images are mapped into common space, and the semantic labels of unseen images are decided by the distances between them and seen images. Although the manuscript is written and organized well, some points should still be noticed.
- What are the differences between your method and the nearest neighborhood?
- The critical issue of zero-shot is knowledge transfer. How do you deal with this problem?
- Do you consider the issue of the domain shift problem?
- The contents of the Related Work section should be enriched.
- The contents of the Relation Module sub-section should be explained in detail. What relations between seen and unseen images can you learn from your method? How do you guarantee the learned relationships are helpful?
- More latest compared methods should be selected to confirm the effectiveness of your model.
Reviewer 2 Report
In computer based image classification, ther problem of a large number of categories without, or too few, pre-labeled training samples appears to be addressed in this paper, which proposes zero-shot learning combined with a learnable "deep metric" to overcome the "semantic gap problem" in the feature mapping process. The new zero-shot image classification method proposed here is combined with an end-to-end learnable deep metric as a relation network where visual features and semantic categories are mapped onto a common workspace. The deep metric is thus exploited for learning similarities between visual features and semantic categories. The paper is reasonably well written apart from minor issues with grammar and spelling that will need to be fixed. The paper contents are abundant in mathematical detail, but lack in sufficiently detailed explanations how the new method permits resolving the "semantic gap problem" in image classification. Major revisions are required to overcome this weakness:
1) The mathematical forms and definitions provided are extensive and we understand that a similarity score is computed on the basis of learnt associations between visual features and semantic categories. However, it is not explained how the "deep metric" deals with problems relative to the inclusivity, exclusivity, and exhaustivity of learnt associations for labeling and classification. Which are the boundary conditions, where are the limitations? How can the "deep metric" be improved further?
2) Although human labeling is often deemed time-consuming and error prone, this is not systematically the case. When shapes and categories are perceptually non-ambiguous, a well-designed algorithm is certainly faster than a human and a "deep metric" as is suggested here could be truly useful. However, when it comes to telling apart highly ambiguous objects (shapes, shades) in complex contexts, human classification is far more reliable. The examples shown here in this paper are images of animals, but surely the method should be fit do deal with other categories? How is the computational transition from top-level categories ("animal" or other) associated with visual features in the image to bottom-level categories ("fox", "cat" or other) represented in the learnable "deep metric"? How generlizable are these computations to categories other than "animal"? Unless I missed something, how do the compuation times here compare with the time that would be necessary for human classification of the same image material?
3) Finally, the authors briefly state in their conclusions that their zero-shot image classification method (based on the suggested learnable "deep metric") overcomes limitations in similarity between visual and semantic features, and therefore "alleviates the semantic gap problem". However, apart from an over-detailed mathematically decription of the computational modules supposed to achieve this, there is no in-depth discussion that explains how exactly the specific properties of the suggested method tackle the various problems associated with the "semantic gap problem". Some of these are pointed out in the points raised here below, to be completed and discussed accordingly, in the introduction and the conclusions.
4) A thorough spell and grammar check of the whole manuscript is required. To give one example: in line 80, the text should read "Related Work" instead of "Relate Work".....
Reviewer 3 Report
The authors proposed an image classification method based on fusing the semantic features with visual features. Several concerns need to be clarified to make the proposed work be convincing.
- The semantic feature v can be in many forms (e.g. annotations, labels, embeddings ). It should be specified which type of semantic features and how they were extracted in this work.
- Considering semantic features with visual features is not a new idea. Similar ideas can be found in this paper: "Multi-Label Zero-Shot Learning via Concept Embedding", https://arxiv.org/abs/1606.00282.
- This work shows weak relations to this journal. The authors should make sufficient reference to related works published in this journal.
Round 2
Reviewer 1 Report
All of the issues have been modified, and the quality of the current version is improved a lot. I suggest that this paper can be accepted.
Reviewer 2 Report
The authors have addressed my queries and comments, and some revisions were performed.